# Vulvar Paget’s Disease: A Systematic Review of the MITO Rare Cancer Group

**DOI:** 10.3390/cancers15061803

**Published:** 2023-03-16

**Authors:** Giuseppe Caruso, Amelia Barcellini, Roberta Mazzeo, Roberta Gallo, Maria Giuseppa Vitale, Anna Passarelli, Giorgia Mangili, Sandro Pignata, Innocenza Palaia

**Affiliations:** 1Department of Maternal and Child Health and Urological Sciences, Sapienza University of Rome, Policlinico Umberto I, 00161 Rome, Italy; 2Department of Experimental Medicine, Sapienza University of Rome, Policlinico Umberto I, 00161 Rome, Italy; 3Radiation Oncology Unit, Clinical Department, CNAO National Center for Oncological Hadrontherapy, 27100 Pavia, Italy; 4Department of Internal Medicine and Medical Therapy, University of Pavia, 27100 Pavia, Italy; 5Department of Medicine (DAME), University of Udine, 33100 Udine, Italy; 6Department of Oncology and Hematology, University Hospital of Modena, 41100 Modena, Italy; 7Department of Urology and Gynecology, Istituto Nazionale Tumori IRCSS, “Fondazione G. Pascale”, 80131 Naples, Italy; 8Obstetrics and Gynecology Unit, IRCCS San Raffaele Scientific Institute, 20132 Milan, Italy

**Keywords:** extramammary Paget’s disease, vulvar Paget’s disease, vulvar cancer, rare gynecological cancers

## Abstract

**Simple Summary:**

Vulvar Paget’s disease (VPD) is an extremely rare malignancy of the vulva with a high local recurrence rate and low mortality. Due to its non-specific symptoms and lack of clinical knowledge, VPD is often misdiagnosed with eczematous skin lesions, and the definitive diagnosis is often delayed. Since currently there is no global consensus on the optimal management of VPD, we present here a systematic review aiming to give readers a concise overview of the state-of-the-art evidence and the emerging therapeutic opportunities.

**Abstract:**

Vulvar Paget’s disease (VPD) is a rare form of cutaneous adenocarcinoma of the vulva, which accounts for about 1–2% of all vulvar neoplasms and mainly affects post-menopausal women. The clinical presentation is usually non-specific and mimics chronic erythematous skin lesions; therefore, the diagnosis is often difficult and delayed. Although VPD is typically diagnosed at a locally advanced stage and has a high recurrence rate, the prognosis is overall favorable with a 5-year survival of nearly 90%. Due to the limited and poor-quality evidence, there is no global consensus on optimal management. Therefore, we performed a systematic review of the literature through the main electronic databases to deepen the current knowledge of this rare disease and discuss the available treatment strategies. Wide surgical excision is recommended as the standard-of-care treatment and should be tailored to the tumor position/extension and the patient’s performance status. The goal is to completely remove the tumor and achieve clear margins, thus reducing the rate of local recurrences. Non-surgical treatments, such as radiotherapy, chemotherapy, and topical approaches, can be considered, especially in the case of unresectable and recurrent disease. In the absence of clear recommendations, the decision-making process should be individualized, also considering the new emerging molecular targets, such as HER2 and PD-L1, which might pave the way for future targeted therapies. The current review aims to raise awareness of this rare disease and encourage international collaboration to collect larger-scale, high-quality evidence and standardize treatment.

## 1. Introduction

Vulvar Paget’s disease (VPD) is the most common extramammary Paget’s disease (EMPD) and accounts for approximately 1–2% of all vulvar neoplasms [1]. EMPD is an extremely uncommon skin malignancy, arising in apocrine gland-rich areas other than the mammary glands and accounting for 6.5% of all Paget’s diseases [2]. It originates from apocrine gland cells and then extends and proliferates within the epithelium [3]. The most common site of origin is the vulva (60–80%), followed by perineal areas (15%) and male genitalia (14%) [4]. Although it is generally limited to the epithelium, in approximately 10% of cases it may progress to invasive disease, metastasizing to local lymph nodes and distant organs [5]. According to Wilkinson and Brown’s classification, EMPD can be distinguished in two different subtypes: (1) primary (cutaneous origin), which originates in the epidermis and can be further classified as (i) in situ or intraepithelial (usual type), (ii) invasive, and (iii) associated with an underlying adenocarcinoma of a skin appendage; (2) secondary (non-cutaneous origin), which is the metastatic spread to the epidermis from (i) anorectal, (ii) urothelial, or (iii) other adenocarcinomas [6].

VPD mainly affects post-menopausal women over the age of 60 years. The clinical presentation is similar to mammary Paget’s disease and typically consists of erythematous or eczema-like chronic skin lesions associated with itchiness, tenderness, burning sensation, and occasionally pain [7]. Lesions are typically multifocal, mostly occur in the labia majora, and appear as well-demarcated patchy erythematous plaques, exhibiting the classic “strawberry and cream” pattern [8]. Given the rarity and non-specific clinical presentation, VPD is often misdiagnosed with dermatitis or eczema, and the correct diagnosis is frequently achieved when the disease is locally advanced [9]. However, with a 5-year overall survival of 75–90% and its indolent behavior, VPD has a favorable prognosis [10]. The initial diagnostic work-up should include a physical examination, staging imaging (pelvic ultrasound, magnetic resonance imaging, and/or computed tomography), and a vulvar excisional biopsy. Additionally, breast examination, coloscopy, cystoscopy, and serum tumor markers may be useful for the differential diagnosis between primary and secondary VPD [11].

Surgery is the mainstay of treatment, both in the primary and recurrent setting [12]. Despite its effectiveness, local recurrences are common, especially in the case of positive surgical margins—even if only pre-invasive disease is present—and multifocal microscopic disease [13]. Several conservative treatments have been described as an alternative to surgery in selected patients and their role needs to be clarified [14].

Currently, there is no global consensus regarding the optimal management of VPD, and thus, this review aims to summarize and critically evaluate the state-of-the-art knowledge of this rare disease providing insights that might be relevant for future treatment strategies.

## 2. Materials and Methods

### 2.1. Search Strategy

A comprehensive literature search was performed up to January 2023 across the following electronic databases: PubMed, EMBASE, Web of Science, Scopus and Cochrane library. The process of evidence acquisition combined the following key words and MESH terms: “vulvar Paget” and “vulvar Paget’s”. Search filters were applied to select clinically relevant articles from databases. The systematic review followed the recommendations of the Preferred Reporting Items for Systematic Reviews and Meta-Analyses (PRISMA). The protocol has not been registered.

### 2.2. Study Selection

In this systematic review, we included the following: (1) full-text English-language articles published in peer-reviewed journals starting from 1995; (2) original articles, including case reports; (3) primary histologically confirmed VPD; (4) description of treatment details and outcomes.

We considered as exclusion criteria the following: (1) primary vulvar tumors other than VPD; (2) secondary VPD; (3) cases with an uncertain diagnosis; (4) synchronous tumors; (5) studies focusing exclusively on histopathological, immunohistochemical, and molecular aspects; (6) studies with different aims than the analysis of treatment and outcome measures; (7) reviews, systematic reviews, meta-analyses, guidelines, books, editorials, letters, comments, conference abstracts, and preliminary studies with animal models.

### 2.3. Data Extraction and Analysis

Three authors (A.B., R.M., A.P.) independently screened article titles, abstracts, and full texts to ensure that all relevant studies were included. Cross-referenced studies identified from searched articles were also evaluated to integrate the literature search. Two authors (G.C. and R.G.) verified the inclusion criteria and excluded irrelevant studies and duplicate records. In the case of overlapping studies, we selected the most recent and/or most comprehensive manuscript. Two authors (I.P. and G.C.) carried out data extraction and quality assessment from all the retrieved studies based on full-text articles. Discrepancies between the investigators were resolved through a consensus. After careful selection of articles suitable for the review, we obtained the following information from each report: authors, year of study publication, study design and setting, number of patients, age, clinical history/presentation, histological and immunohistochemical features, tumor stage, molecular profile, type of treatment, adverse events, follow-up, disease control rate (DCR), recurrence rate, site of recurrence, progression-free survival (PFS), and overall survival (OS). Data were reported according to the Preferred Reporting Items for Systematic Reviews and Meta-Analyses (PRISMA) [15].

### 2.4. Statistical Analysis

Before data analysis, we performed an exploration phase of the data. The categorical data were described by frequency and percentage, with continuous data by mean, median, and range. If necessary, for the description of the endpoints, the percentages related to the number of patients or studies for which those specific data were available were reported. All analyses were performed using SPSS statistical software, version 20.0 (SPSS Statistics; International Business Machines Corporation [IBM], Armonk, NY, USA) for Mac.

## 3. Results

### 3.1. Study Selection

Figure 1 shows the flowchart of the systematic literature search process. The study selection resulted in a total of 281 relevant articles. Another 16 articles were identified via cross-referencing and hand-searching bibliographies. Through a process of screening, 96 studies met the inclusion/exclusion criteria, all with non-overlapping patients.

### 3.2. Study and Population Characteristics

The main characteristics of the included studies are detailed in Table 1. Of the 96 studies included, 5 were prospective, 24 were retrospective, 30 were case series, and 37 were case reports. No randomized controlled studies were found. The 96 selected studies involved a total of 5617 VPD patients, and among all studies, the sample size ranged from one patient in case reports to 2602 patients in larger series. The age at diagnosis across all studies ranged between 29 and 100 years (mean: 71 years). In 63–100% of the cases, the main sign at diagnosis was an erythematous/eczematoid lesion of the vulva, and persistent vulvar itching was the most commonly reported symptom (8–100%). The less common symptoms/signs were vulvar burning (7.2–70.8%), vulvar pain (5.9–45.8%), vaginal discharge (4–13.9%), and bleeding (2–16.3%). Approximately 7–30% of reported cases were micro-invasive, while 5–67.8% were invasive.

### 3.3. Treatment Approches

Surgery was the first treatment choice in most studies (75%). The main surgical approach consisted in a wide local excision (10–96.7%), followed by partial simple vulvectomy (2.9–88%), radical vulvectomy (4.2–58%), total simple vulvectomy (2–62.5%), hemivulvectomy (9.7–28%), and skinning vulvectomy (1–37.5%). According to the reports, the margin status was positive in the range of 7.1% and 91.7%. In the case of positive margins and/or positive lymph nodes, patients underwent external beam radiotherapy (RT) with a total dose ranging between 23 Gy and 61 Gy. For unresectable disease, radical external beam RT was used up to a total dose of 63 Gy. Among RT approaches, Boron Neutron Capture Therapy (BNCT) was delivered in a radical setting. Systemic chemotherapy was almost exclusively used in the case of metastatic disease with palliative intent and very rarely in the neoadjuvant and adjuvant settings. The most commonly reported antineoplastic agents included bleomycin, mitomycin, 5-fluorouracil, cisplatin or carboplatin, paclitaxel or docetaxel, and trastuzumab. Local approaches included topical chemotherapy with imiquimod 5%, laser therapy, and photodynamic therapy using methyl 5-aminolevulinic acid or hematoporphyrin derivatives as a photosensitizer. Details on the treatment strategies are described in Table 2.

### 3.4. Outcomes

When reported, the outcome measures varied widely across the included studies. The pooled median follow-up ranged between one month and 9 years. The disease control rate was achieved in 50–100% of patients. The recurrence rate ranged between 23% and 73%, with a mean time to recurrence of 1–4.4 years. The progression-free survival rates ranged between 0.5 and 5 years, and the overall-free survival ranged between one and 8 years. No significant postoperative complications or treatment-related toxicities higher than grade 3 were reported. Where available, data on the type of treatment at recurrences are outlined in Table 3. In cases of local recurrence, surgical excision remained the most frequent treatment choice, followed by topical approaches (using imiquimod and 5-fluorouracil) and RT. Systemic chemotherapy was mainly used in the case of distant progressive disease. Given the high heterogeneity, it is difficult to provide an exhaustive summary and draw definitive conclusions. Details on the outcomes of each study are shown in Table 3.

## 4. Discussion

Due to the rarity and absence of standardized guidelines, the management of VPD poses a huge challenge in clinical practice [9,14,111]. In the present systematic review, we collected the best available evidence on this rare disease and analyzed the clinical decision-making process, treatment, and outcomes of 5617 VPD patients from 96 different studies. Given the high heterogeneity of the studies and population characteristics, a meta-analysis was not feasible.

This review confirmed that VPD patients are mostly postmenopausal women with a mean age of approximately 70 years, presenting with an itching, persistent erythematous lesion of the vulva. The final diagnosis of VPD was achieved late in most of the cases that were treated in non-referral centers. The diagnostic pathway and treatment strategies varied widely across the analyzed studies. Surgery was the first choice for the primary treatment of VPD with a recurrence rate of 20–70%. Depending on the site and extent of the lesion, the surgical approaches ranged from wide local excision to radical vulvectomy with a reconstruction flap. To date, there is no consensus on which surgical technique minimizes local recurrence. There is, therefore, the need to better define novel surgical prognostic factors and promote global standardization to minimize long-term morbidity and improve patient care and quality of life [112]. Preoperative vulvovaginal intensive biopsy mapping (called “clock mapping”), both inside and outside visible lesions, became a useful workup tool for predicting the invasiveness and extension of VPD and tailoring the radicality of surgery [113]. Inguinofemoral lymphadenectomy (or sentinel lymph node biopsy) should be performed in invasive diseases. Re-excision can be considered in cases of positive resection margins. Due to the multifocality and irregular shape of VPD lesions, the surgical margins are often positive and local relapses are frequent. Surgery was the preferred option to treat local recurrences, even if adjuvant RT could be delivered in the case of positive margins, dermal invasion, or lymph node metastasis.

The role of non-surgical therapies, such as RT, topical imiquimod, chemotherapy, photodynamic therapy, and laser CO_2_ therapy, remains unclear and can be considered in cases of unresectable, recurrent, and metastatic disease, as well as a valid conservative alternative for non-invasive disease [5,114,115]. It is noteworthy that preliminary results indicated that noninvasive physical plasma might be a viable treatment option for women with a cervical intraepithelial neoplasia, and this approach might be explored for VPD patients as well [116]. The role of RT as an option in the treatment of VPD has not been fully understood. As RT appeared unsatisfactory in the adjuvant setting, further investigation into the potential role of combining RT with immunotherapy is warranted. The role of radical RT in elderly patients with surgical contraindications or as an alternative to surgery in the recurrent setting remains unclear. However, it should be stressed that, across the selected studies, the technical details on RT were rarely reported and, where available, the protocols varied widely across the reports, underlining the need to standardize the doses and volumes in future multicenter studies. Moreover, there were no reports on particle beam RT in this challenging scenario, and considering the ballistic and radiobiological advantages of this innovative technique, this approach should be tested in the future to reduce toxicities and improve the total dose to the target. Moreover, even if systemic therapy should be recommended for metastatic VPD patients, further research is needed on the role of chemotherapy in the neoadjuvant and adjuvant settings.

Overall, the prognosis of VPD patients is favorable. Natural history shows high rates (50–100%) of disease control at primary diagnosis. However, the rate of local recurrences is also high, and repeated surgical procedures can be mutilating and impair quality of life. Therefore, in the era of precision medicine, novel targeted therapies are urgently needed. Owing to the rarity and lack of high-level evidence on VPD, standardized follow-up protocols are also lacking. The follow-up should consider regular vulvar inspection, vulvoscopy, re-biopsy in suspicious cases, and CT/MRI in cases of distant lesions.

There were lacking data on the potential identification of molecular targets. The strong heterogeneity found in the histopathological and immunophenotypic reports highlights the importance of promoting centralized histological reviews of these rare specimens to ensure a correct diagnosis and standardize the management and patient care. Indeed, determining the expression of hormone receptors, as well as the human epidermal growth factor receptor 2 (HER2) or the programmed death-ligand 1 (PD-L1) status, is crucial to paving the way for personalized targeted therapy, especially in cases of advanced/metastatic disease or recurrence [117]. In a recent systematic review and meta-analysis including patients with extramammary Paget’s disease, the expression rates of hormones receptors were 12% (95% CI = 0.03–0.36) for estrogen receptors (ERs), 9% (95% CI = 0.03–0.25) for progesterone receptors (PRs), and 40% (95% CI = 0.34–0.47) for androgen receptors (ARs) [118]. This appears in contrast with results of Garganese et al. who reported the following expression rates: 70% ER, 20% PR, and 75% AR positivities among 41 patients with VPD [119]. Of note, no significant differences in terms of hormone expression were seen between invasive and non-invasive diseases. Anti-hormonal targeted therapy represents an interesting treatment option worth further investigation. HER2 overexpression was found in 32% of women with extramammary Paget’s disease [118], and its status needs to be assessed due to its pathogenetic role and correlation with nodal metastases, local invasion, and recurrence rates. However, more data regarding the HER2 status in VPD are required before drawing definitive conclusions about the potential use of biological therapies targeting HER2. Moreover, still, too little is known about the PD-L1 and tumor-infiltrating lymphocyte (TIL) status in vulvar Paget’s disease. Data on PD-L1 expression were first reported by Garganese et al. based on a cohort of 41 patients (10% non-invasive VPD and 27% invasive VPD) and are interesting as they might open up the possibility of using immunotherapy, either alone or in combination with RT [119,120].

The major limitations of this review are the small sample sizes and the long study periods (from 1995 to 2022), which could reflect several changes in the histological diagnosis and treatment management (surgical and RT technologies, as well as available chemotherapy schedules). Therefore, no definitive conclusions on the impact of these treatments on outcomes can be drawn. Moreover, the heterogeneity of the analyzed population, missing data, the study design (mainly case reports or case series), and the absence of controlled arms represent other relevant limitations. On the other hand, phase III randomized studies are not feasible and an evidence-based approach for these rare tumors is not so easy. Oncological prospective registers and enrollment in basket clinical trials, including VPD, are warranted, especially in the advanced/metastatic and recurrent setting. Despite the heterogeneity, our data contribute to the limited literature evidence by drawing more attention to VPD patients and highlighting the need of optimizing the clinical decision-making process and standard of care. 

## 5. Conclusions

In the present systematic review, we summarized the state-of-the-art literature evidence on the vulvar Paget’s disease, with a focus on available treatment approaches and outcomes. Due to the rarity of the disease, the low-quality evidence (mostly small retrospective studies), and the huge heterogeneity in terms of reported treatment strategies and outcome measures, it is not possible to obtain clear recommendations on the best management. Our findings emphasize that VPD patients should be centralized in referral centers and managed through high-skilled collaborative networks, with a multidisciplinary approach where the treatment strategy is discussed on a case-by-case basis. The centralization of care for rare tumors has already proved to significantly increase patient outcomes. Strong inter- and multidisciplinary collaborations are crucial to create networks, sharing data and comparing different single-institution experiences. International databases are urgently needed as they may lead to a real step forward in understanding such a challenging disease.

## Figures and Tables

**Figure 1 cancers-15-01803-f001:**
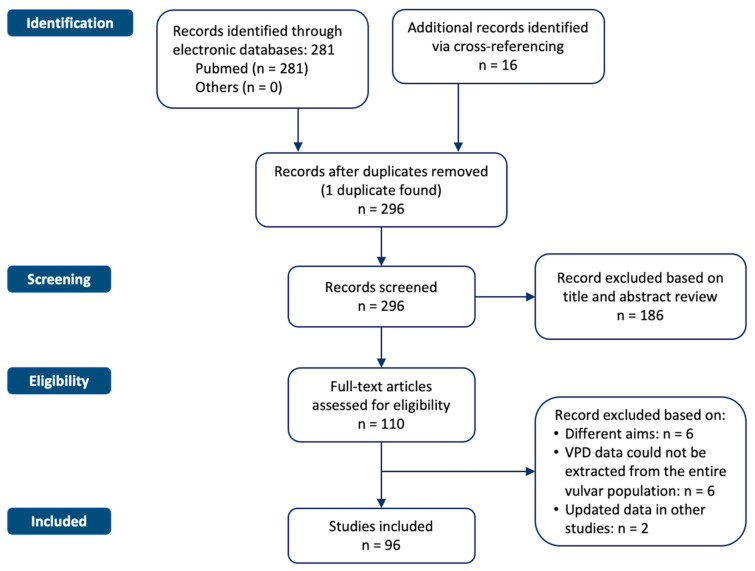
Flowchart of the systematic literature search process. The study selection resulted in a total of 281 relevant articles.

**Table 1 cancers-15-01803-t001:** Main characteristics of the included studies.

Reference	Type of Study	Sample Size, n	Mean Age (Years, Range)	Symptoms (%)	Signs (%)	Tumor Invasion
Kodama et al. 1995 [16]	Cs	30	67.6 (41–82)	Vulvar itching (100%)	Erythema (63%)	10 ni-VPD (33.3%); 9 mi-VPD (30%); 11 i-VPD (36.7%)
Fishman et al. 1995 [17]	Cs	14	70.5 (57–83)	Vulvar itching (100%)	NR	NR
Yoshitatsu et al. 1997 [18]	Cr	1	68	Vulvar itching	Vulvar erythema with a nodule on the left labium major	ni-VPD
Goldblum et al. 1997 [19]	Cs	19	65 (56–86)	NR	NR	5 i-VPD (26.3%)
Fanning et al. 1999 [20]	Ra	100	70 (35–100)	Vulvar itching (NR)	NR	7 mi-VPD (7%); 5 i-VPD (5%)
Henta et al. 1999 [21]	Cr	1	74	Pain	Irregular border dark red mass	ni-VPD
Murata et al. 1999 [22]	Cs	1 (extracted from a total of 6 pts)	78	NR	Erythema	i-VPD
Louis-Sylvestre et al. 2001 [23]	Ra	52	67.4 (39.6–94.7)	NR	NR	NR
Wilkinson et al. 2002 [6]	Cs	3	76 (76–81)	Vulvar inflammatory lesion (66%); urinary urgency (33%)	Erythematous and eczematoid lesion (66%); periurethral induration and scar tissue surrounding the urethral meatus (33%)	NR
Tebes et al. 2002 [24]	Cs	23	69(46–84)	Itching (78%), burning sensation (52%), dysuria (4%), watery discharge (4%)	Vulvar lesion (87%)	6 i-VPD (26.1%)
Luk et al. 2003 [25]	Cs	1 (extracted from a total of 6 pts)	55	NR	NR	NR
Wang et al. 2003 [26]	Cr	1	75	Itching and burning	Hypopigmented to pink plaque in labium major	NR
Chin et al. 2004 [27]	Cr	1	65	NR	NR	NR
Zawislak et al. 2004 [28]	Cr	1	66	8 years of persistent vulvar itching	Purple-red, well-demarcated, moist lesion on the left labia majora	NR
Bhattacharya et al. 2005 [29]	Cr	1	59	vulvar irritation	NR	ni-VPD
Raspagliesi et al. 2006 [30]	Prospective pilot study	7	63 (50–75)	Severe vulvar itching (NR)	NR	NR
Yanagi et al. 2007 [31]	Cr	2	84 (82–86)	NR	Infiltrated reddish plaque on the genital region	NR
Hatch et al. 2008 [32]	Cs	2	64 (60–68)	Pain (100%), urinary retention (50%)	Vulvar erythema	NR
Karam et al. 2008 [33]	Cr	1	34	NR	NR	NR
Challenor et al. 2009 [34]	Cs	2	57 (48–66)	Itching (100%), soreness (50%)	Vulvar erythema	NR
Sendagorta et al. 2010 [35]	Cs	3	66 (58–82)	Itching (66.6%), burning (33.3%)	Vulvar lichenified plaque (33.3%), vulvar erythematous plaque (66.6%), vulvar erosions (1–33.3%)	NR
Shaco-Levy et al. 2010 [36]	Cs	56	69 (42–89)	Vulvar itching (91%), vulvar pain (11%), drainage (5%), bleeding (2%)	Erythematous-white plaque (98%)	0.2–6 mm
Roh et al. 2010 [37]	Cs	11	66.6	Pruritis	NR	NR
Anton et al. 2011 [38]	Cr	1	84	Itching	Hyper-pigmented to the pink plaque from the pubic area to the left great labia	NR
Feldmeyer et al. 2011 [39]	Cr	1	59	Fluctuating soreness and itching	Poorly delimited erythematous–squamous patch on the right labia major	NR
Hanawa et al. 2011 [40]	Cr	1	70	Asymptomatic	Increasing vulvar erythema, slightly raised erythematous lesion with ulcers, hemorrhagic granulomatous lesion	NR
Jones et al. 2011 [41]	Ra	50	67.6	Itching (54%)	NR	NR
Tonguc et al. 2011 [42]	Cr	1	65	Itching	Hyperkeratotic erythematous lesion from right labium to the clitoris	NR
Mendivil et al. 2012 [43]	Cs	16	75 (53–84)	Pruritis (100%)	NR	3 i-VPD
Al Yousef et al. 2012 [44]	Cr	1	78	Pruritis	Erythema	NR
Baiocchi et al. 2012 [45]	Cs	4	62.2 (56–80)	Itching (100%)	Vulvar erythematous plaque (100%), vulvar erosions (100%)	NR
Wakabayashi et al. 2012 [46]	Cr	1	68	NR	NR	NR
Cai et al. 2013 [47]	Ra	43	68.6 (52–85)	Itching (95.3%), vulvar pain (18.6%), bleeding (16.3%), and discharge (13.9%)	Erythematous lesions (81.4%), ulceration (32.6%), erosion (30.2%)	NR
Choi et al. 2013 [48]	Cs	3 (extracted from a total of 10 pts)	73 (65–81)	NR	Vulvar erythema	NR
Gavriilidis et al. 2013 [49]	Cr	1	75	Rash and vulvar itching	Eczematoid and erythematous area of the genitalia	NR
Sanderson et al. 2013 [50]	Cs	6	71.5 (58–85)	Soreness (50%), itching (50%), irritation (16,6%), inflammation (16.6%) pain (16.6%)	Plaque	NR
De Magnis et al. 2013 [51]	Ra	34	68.7	Itching (76.5%), burning (58.8%)	NR	i-VPD: 11.7%
Treglia et al. 2013 [52]	Cr	1	61	NR	NR	NR
Magnano et al. 2013 [53]	Cr	1	84	Slight pruritus	Erythematous ulcerated and well-demarcated oval plaque (5 × 3 cm)	ni-VPD
Marchitelli et al. 2014 [54]	Cs	10	71.9 (60–92)	NR	Erythematous plaque	NR
Liu et al. 2014 [55]	Ra	85	64.4 (33–82)	Pruritis (74.1%), pain (5.9%)	NR	i-VPD: 20%
Luyten et al. 2014 [56]	Ra	20	66.4 (41–84)	NR	NR	NR
Carrozzo et al. 2014 [57]	Cs	4 (based on a total of 5 pts)	69 (59–79)	NR	Case 1: itchy erythematous lesion Case 2: itchy dermatosis in the vulvar region+ erythematous–eczematoid lesion in the inferior perivulvar region Case 3: itchy dermatosis in the perivulvar region Case 4: itchy dermatosis in the perivulvar region	NR
Frances et al. 2014 [58]	Cr	1	80	NR	Itching eczematous plaques	ni-VPD
Hata et al., et a. 2015 [59]	Ra	21	69 (53–88)	NR	NR	NR
Asaka et al. 2015 [60]	Cr	1	80	Vulvar itching, pain	Pigmented vulvar, perianal erythematous plague, and a subcutaneous nodule in the left major labia	Intradermal invasion
Sopracordevole et al. 2016 [61]	Ra	27	66.5 (36–88)	Itching (64.3%), burning + itching (14.3%), pain (14.3%), burning (7.2%)	Erythema (93.7%), hyperkeratosis (31.6%)	8 mi-VPD (29.6%) 3 i-VPD (11.1%)
Cowan et al. 2016 [62]	Prospective pilot study	8	71.5 (47–78)	Itching (63%), burning (25%), pain (25%)	Visible lesions (100%), erythema (75%), anal involvement (13%)	NR
Fan et al. 2016 [63]	Ra	18	65	Pruritus, erosions, burning sensation, pain	Erythematous patches (50%), hypopigmentation (50%), hyperpigmentation (27.8%)	NR
Nagai et al. 2016 [64]	Cs	2	75 (69–81)	Itching	Eczema	NR
Onaiwu et al. 2016 [65]	Ra	89	67(32–89)	Pruritus (48.3%)	NR	7 i-VPD (7.9%)
Hillmann et al. 2016 [66]	Cr	1	48	Itching	Acetowhite areas in left minora and majora labia, perineal, and anterior perianal regions	NR
Liau et al. 2016 [67]	Ra	3	74.7 (66–83)	Pruritus, pain	Erythematous patch or plaque	NR
Jeon et al. 2016 [68]	Cr	1	71	Pruritus, pain	Erythematous plaque on both labia majora	ni-VPD
Dogan et al. 2017 [69]	Cr	1	73	Itching	Left vulvar erythema	Intradermal pagetoid cells
Vicentini et al. 2017 [70]	Cr	1	62	Itching, pain, discomfort	Erythematous plaque	ni-VPD
Parashurama et al. 2017 [71]	Cs	18	76.9	Vulvar itching, soreness	Ulcer, mass, pigmented lesion	NR
Konstantinova et al. 2017 [72]	Cs	181	65	NR	NR	NR
Mota et al. 2017 [73]	Cr	1	78	Pruritus	Cutaneous plaque with eczema	NR
Long et al. 2017 [74]	Cs	90	70.3	NR	NR	NR
Kato et al. 2018 [75]	Cr	2	72.5 (64–81)	Nodules and skin infiltrations	Both inguinal lymph nodes	ni-VPD
Borghi et al. 2018 [76]	Ra	79	67.5 (53–77)	NR	NR	mi-VPD/i-VPD: 12.7%
Sawada et al. 2018 [77]	Ra	9	82 (61–94)	NR	NR	ni-VPD
Hiratsuka et al. 2018 [78]	Cs	1 (extracted from a total of 4 pts)	69	NR	NR	i-VPD
Hsieh et al. 2018 [79]	Cr	1	54	NR	NR	NR
Nitecki et al. 2018 [80]	Cs	44	67	Pain (10–22%), pruritis (10–22%), pain + pruritis (6–13%)	NR	i-VPD: 20%
Rioli et al. 2018 [81]	Ra	13	70 (52–84)	NR	NR	3%
Kato et al. 2018 [82]	Ra	17 (7 women)	76.9	NR	NR	i-VPD
Cai et al. 2018 [83]	Cs	8	64.4	NR	NR	NR
Bouceiro-Mendes et al. 2019 [84]	Cr	1	72	Vulvar itching	Erythematous plaque	NR
Loiacono et al. 2019 [85]	Cs	24	69.3 (38–84)	Itching + burning + pruritus + vulvar lesions (21%), pain + pruritus (13%), vulvar itching (4%), pruritus (4%), unknown (58%)	NR	NR
Mujukian et al. 2019 [86]	Cr	1	84	NR	Large erythematous plaque affecting the right labia majora	NR
Molina et al. 2019 [87]	Cs	3	50 (50–70)	Fatigue, weakness, pruritus, fever	Erythematous plaque, scattered, pink-red plaques	NR
Hirai et al. 2019 [88]	Cs	5	65 (53–74)	NR	NR	NR
van der Linden et al. 2019 [89]	Ra	113	73 (41–97)	NR	NR	ni-VPD: 77% mi-VPD: 8.8% i-VPD: 14.2%
Panoskaltsis et al. 2019 [90]	Cr	1	75	Itching, irritation, and burning	Erythematous plaque with white scaling	i-VPD
Nasioudis et al. 2020 [91]	Ra	2602	72 (31–90)	NR	NR	i-VPD: 61.8%
Bartoletti et al. 2020 [92]	Cs	4	62.5 (45–74)	Vulvar itching	NR	NR
Bruce et al. 2020 [93]	Cr	1	37	Vulvar irritation at 7 weeks gestation with an erythematous, edematous left labia	NR	NR
Noel et al. 2020 [94]	Cr	1	65	Vulvar itching	Vulvar reddish lesion	NR
Kilts et al. 2020 [95]	Ra	1268	73 (29–100)	NR	NR	NR
Rathore et al. 2020 [96]	Cr	1	59	Vulvar itching	Vulvar redness, erythematous indurated plaque with multiple papules and nodules	NR
Sarkar et al. 2020 [97]	Cr	1	51	Vulvar itching for 3 years	Erythematous patchy lesion	NR
Sopracordevole et al. 2020 [98]	Ra	10	65.5 (40–80)	Vulvar itching	Red asymmetrical areas, ulcerations, hyperkeratotic red areas, whitish areas	mi-VPD
Stasenko et al. 2020 [99]	Ra	26	62 (29–77)	NR	NR	mi-VPD
Liang et al. 2021 [100]	Cs	38	64.4 (39–79)	Vulvar itching and pain	NR	NR
Hirata et al. 2021 [101]	Cr	1	55	Recurrent fever for 10 months, inguinal and intraperitoneal lymphadenopathies	NR	NR
Kosmidis et al. 2021 [102]	Cr	2	75 (69–81)	Vulvar itching and burning sensation; perineal and vulvar itching and swelling	Erythematous plaque Eczematous plaque + erosion	ni-VPD
Mazzilli et al. 2021 [103]	Cr	1	60	NR	NR	NR
Preti et al. 2021 [104]	Ra	122	65 (36–92)	Itching (59–61%), burning sensation (18%), itching + burning (20–21%)	NR	mi-VPD: 21% i-VPD: 16.8%
Liu et al. 2021 [105]	Ra	54	72 ± 7 (SD)	NR	NR	NR
Ferrara et al. 2021 [106]	Prospective study	10	79 (67–92)	NR	NR	NR
Bajracharya et al. 2022 [107]	Cr	1	70	Vulvar itching and a gradually progressive reddish lesion	NR	NR
Wang et al. 2022 [108]	Prospective pilot study	2 (extracted from a total of 11 pts)	73 (67–79)	NR	NR	NR
Borella et al. 2022 [109]	Ra	55	63 (36–92)	Itching (29–59%), burning (15–27%)	NR	NR
Van der Linden et al. 2022 [110]	Prospective study	24	67 (42–84)	Itching (83%), pain (45.8%), burning sensation (70.8%), strangury (29.2%), dyspareunia (38.5%)	Erythema (100%), scaling (62.5%), ulceration (25%)	NR

AWD = alive with disease; Cr = case report; CR = complete response; Cs = case series; CT = chemotherapy; DCR = disease control rate; DID = died of intercurrent disease; DOD = died of disease; HV = hemivulvectomy; i-VPD = invasive vulvar Paget’s disease; LA = lymphadenectomy; LFU = lost to follow-up; LN = lymph node; MAL-PDT = methyl 5-aminolevulinic photodynamic therapy; mi-VPD = microinvasive vulvar Paget’s disease; NED = no evidence of disease; ni-VPD = noninvasive vulvar Paget’s disease; NR = not reported; PD = Progressive disease; PR = partial response; pts = patients; PV = simple partial vulvectomy; R = residual disease; Ra = retrospective analysis; RT = radiotherapy; RV = radical vulvectomy; SD, standard deviation; SV = simple vulvectomy; TZB = trastuzumab; Vu = vulvectomy; w = weekly; WLE = wide local excision.

**Table 2 cancers-15-01803-t002:** Overview of treatment approaches.

Reference	Surgery			Radiotherapy			Systemic Chemotherapy			Other Therapy
	N	Technique Margin Status	Complications	N	Aim Total Dose/Fr Technique Volumes	Toxicity	N	Aim Scheme	Toxicity	N
Kodama et al. 1995 [16]	29	ni-VPD: RV + bilateral LA + skin graft (30%), RV + bilateral LA (20%), SV (30%), SV + skin graft (10%), RV + bilateral LA + skin graft + perineal resection and artificial anus (10%) Margin status: R+ in 4 cases (40%) mi-VPD: RV + bilateral LA + skin graft (44.4%), SV (33.3%), RV + bilateral LA (22.2%) Margin status: R+ in 5 cases (55.6%) i-VPD: RV + bilateral LA + skin graft (27.3%), RV + bilateral LA (27.3%), RV + bilateral LA + skin graft + total hysterectomy (9%), exploratory laparotomy (9%), resection (9%), RV + bilateral LA + skin graft + perineal resection and artificial anus (9%) Margin status: R+ in 4 cases (36.4%)	NR	3	NR	NR	11	Aim: palliative Scheme: bleomycin, mitomycin, FCAP (5-FU, cyclophosphamide doxorubicin, cisplatinum)	NR	NR
Fishman et al. 1995 [17]	14	Primary surgery: WLE (57.1%), SV (21.4%), RV (21.4%) Recurrent surgery: WLE (78.6%), SV (7.1%), WLE + bilateral LA + rectus abdominal myocutaneous reconstruction (7.1%) Margin status: R+ in 5 cases (35.7%)	NR	1	Aim: adjuvant Total dose/Fr: NR Volumes: vulvar, pelvic and groin RT	NR	0	-	-	0
Yoshitatsu et al. 1997 [18]	1	Lesion resection + LA + gracilis musculocutaneous flap → WLE + split-thickness skin graft Margin status: NR	NR	0	-	-	1	Aim: adjuvant Scheme: mitomycin C and Tegafur	NR	0
Goldblum et al. 1997 [19]	19	ni-VPD: SV (10), RV (2), RV + LA (1) Margin status: NR i-VPD: RV (3), SV (2), RV + LA (1) Margin status: R+	NR	0	-	-	0	-	-	0
Fanning et al. 1999 [20]	100	WLE (32%) RV (58%) HV (10%) Margin status: NR	NR	0	-	-	0	-	-	0
Henta et al. 1999 [21]	0	-	-	1	Aim: palliative, electron beam irradiation Total dose/Fr: 5000 cGy Volumes: vulvar lesion	NR	1	Aim: palliative Scheme: etoposide (VP16) 100 mg for 5 days concomitant with RT	NR	1 (photodynamic therapy with aminolevulinic acid)
Murata et al. 1999 [22]	0	-	-	1	NR	NR	0	-	-	0
Louis-Sylvestre et al. 2001 [23]	31	WLE (67.7%) Vu (22.6%) HV (9.7%) Margin status: R+ in 6 cases of Vu	NR	0	-	-	0	-	-	21
Wilkinson et al. 2002 [6]	3	Case 1: RV Case 2: RV, with excision to the fascia and partial vaginal excision, including excision of the hymen Case 3: 2 subsequent partial Vus ➝ periurethral scar tissue excision Margin status: Case 1: R0 Case 2: R0 Case 3: NR	NR	0	-	-	0	-	-	0
Tebes et al. 2002 [24]	23	WLE o PV (73.9%) Radical resection + LA (26.1%) Margin status: R+ in 13 cases (50%)	NR	0	-	-	0	-	-	0
Luk et al. 2003 [25]	1	RV + LA Margin status: NR	NR	1	Aim: adjuvant Total dose/Fr: 60 Gy⁄30 fr⁄6 weeks to primary site and right inguinal nodal site 32 Gy⁄8 fr⁄2 weeks at 90% isodose level for prophylactic left inguinal nodal RT Volumes: anteroposterior opposed photon fields to pelvis and vulva, then electron boost to right vulva, matched 10 MeV electron fields to bilateral inguinal nodal sites	Confluent wet desquamation, resolved at 4 weeks post-RT, mild acute small bowel irritation	0	-	-	0
Wang et al. 2003 [26]	1	WLE Margin status: NR	NR	0	-	-	0	-	-	1
Chin et al. 2004 [27]	1	SV → WLE + rectus abdominal flap Margin status: NR	NR	0	-	-	0	-	-	0
Zawislak et al. 2004 [28]	0	-	-	0	-	-	0	-	-	1 (photodynamic therapy)
Bhattacharya et al. 2005 [29]	1	PV → 4 excisions → hysterectomy + salpingo-oophorectomy + LA + SV + partial vaginectomy Margin status: NR	NR	0	-	-	0	-	-	0
Raspagliesi et al. 2006 [30]	5	Vu or WLE Margin status: R+	NR	0	-	-	0	-	-	2
Yanagi et al. 2007 [31]	0	-	-	2	Aim: radical Total dose/Fr: electron beam of 4 MeV, 2.25 Gy per day, 4 days/week to a total dose of 45 Gy. Volumes: a 10 mm margin surrounding the lesion Aim: radical Total dose/Fr: X-rays of 4 MV, 2.5 Gy per day, 4 days/week to a total dose of 45 Gy + electron beam therapy (4 MeV, 3.0 Gy per day, for 5 fractions to a total dose of 15 Gy) Volumes: thick lesions on both sides of the genitals	mild acute mucositis (2)	0	-	-	0
Hatch et al. 2008 [32]	2	Local resection (1), RV (1) Margin status: NR	NR	0	-	-	0	-	-	2
Karam et al. 2008 [33]	1	Surgical excisions of the vulva and perineum (8 times) Margin status: NR	NR	-	-	-	1	Aim: palliative Scheme: 3-w-TZB (14 cycles)	None	0
Challenor et al. 2009 [34]	2	Skinning Vu and reconstruction with split skin graft (1), skinning Vu with reconstruction via V-Y advancement flaps (1) Margin status: R+	NR	0	-	-	0	-	-	2
Sendagorta et al. 2010 [35]	1	Skinning posterior vulvectomy with reconstruction (1)	NR	0	-	-	0	-	-	3
Shaco-Levy et al. 2010 [36]	56	Conservative procedures: WLE, SV, PV. Radical procedures: complete Vus and RVs Margin status: R1 in 44%	Skin flap/graft breakdown and necrosis (6); wound infection (2); hypertrophic scar (2); thigh cellulitis (1)	6	Aim: adjuvant Details on RT available only for 1 case Total dose/Fr: 2340 cGy to the groins, pelvis, and anus, with a 4500 cGy boost to the groins and anus and 6100 cGy electron boost to the peri-anal region	NR	0	-	-	0
Roh et al. 2010 [37]	11	WLE: 63.6% RV: 18.1% SV: 18.1% Margin status: R+ in 72.7%	-	0	.	.	0	-	0	-
Anton et al. 2011 [38]	0	-	-	0	-	-	0	-	-	1
Feldmeyer et al. 2011 [39]	0	-	-	0	-	-	0	-	-	1
Hanawa et al. 2011 [40]	0	-	-	-	Aim: palliative Total dose/Fr: NR Technique: NR Volumes: NR	NR	-	Aim: palliative Scheme: w-TZB for 5 weeks ➝ w-paclitaxel plus w-TZB 6 cycles	NR	0
Jones et al. 2011 [41]	50	WLE: 24% HV: 28% RV: 28% Anterior vulvectomy: 14% Skinning vulvectomy: 2% SV: 2% Margin status: R+ in 54%	-	0	-	-	2	NR	-	0
Tonguc et al. 2011 [42]	1	WLE → WLE Margin status: R+ → R0	NR	0	-	-	0	-	-	1
Mendivil et al. 2012 [43]	16	SV: 62.5% RV: 12.5 % Only biopsy: 12.5% Margin status: R+ in 68.8%	NR	0	-	-	0	-	-	-
Al Yousef et al. 2012 [44]	0	-	-	0	-	-	0	-	-	1
Baiocchi et al. 2012 [45]	2	SV (1), WLE (1) Margin status: R+	NR	1	Aim: adjuvant Total dose/Fr: 54 Gy Technique: external RT Volumes NR	NR				4
Wakabayashi et al. 2012 [46]	1	WLE and radical hysterectomy ➝ resection of recurrence in the right pelvic floor Margin status: NR	NR	0	-	-	1	Aim: palliative Scheme: 3-w-TZB (17 cycles)	Moderate headache and flushing during the first infusion, no other adverse effects	0
Cai et al. 2013 [47]	35	(1) Radical surgery (RV +/− inguinal LA): EMPDV + adnexal adenocarcinoma (100%), intraepithelial EMPDV (44%), i-VPD (42.8%); (2) Conservative surgery (WLE and SV): i-VPD (57.2%), intraepithelial EMPDV (56%) Margin status: R+: 45.8% in intraepithelial EMPDV, 42.8% in i-VPD and 66.7% in EMPDV + adnexal adenocarcinoma	NR	17	Median total dose/fr: primary radical RT (8): median total dose 60 Gy/20 fr (60–63 Gy)/20–29 fr); Adjuvant RT after surgery (6): median total dose 57 Gy (range = 39–60 Gy); Preoperative RT (1): total dose 43.30 Gy; Rescue RT (1): 63 Gy Palliative RT in tibial metastasis (1): NR Technique: Co^60^ or 9 MeV electrons Volume: pelvic, para-aortic, vulvar, inguinal, tibial	NR	1	Aim: neoadjuvant before surgery Scheme: 2 cycles of unspecified CT	NR	0
Choi et al. 2013 [48]	3	WLE Margin status: NR	NR	0	-	-	0	-	-	3 (topical imiquimod)
Gavriilidis et al. 2013 [49]	1	RV with bilateral inguinal LN dissection Margin status: R0	NR	0	-	-	0	-	-	0
Sanderson et al. 2013 [50]	2	Vu (1), skinning posterior Vu with reconstruction (1)	NR	0	-	-	0	-	-	6 (topical imiquimod)
De Magnis et al. 2013 [51]	34	WLE (61.7%) Partial SV (29.4%) RV (5.9%) Total SV (2.9%) Margin status: R+ in 44.1%	NR	0	-	-	0	-	-	0
Treglia et al. 2013 [52]	1	Radical excision Margin status: R-	NR	0	-	-	0	-	-	0
Magnano et al. 2013 [53]	0	-	-	0	-	-	0	-	-	Photodynamic therapy with aminolevulinic acid
Marchitelli et al. 2014 [54]	0	-	-	0	-	-	0	-	-	10
Liu et al. 2014 [55]	69	WLE (17.4%), PV (2.9%), SV (37.6%), RV (34.7%) Margin status: R+ in 34.5%	NR	4	NR	-	0	-	-	0
Luyten et al. 2014 [56]	0	-	-	0	-	-	0	-	-	20
Carrozzo et al. 2014 [57]	0	-	-	4	Radical HDR brachytherapy with 188Re (Dermo-Beta-Brachytherapy, DBBT) - first course (4 pts; 50 Gy for 300 µ) - second course (3 pts; 50 Gy for 300 µ)	Burning sensation and superficial erosions, followed by small crusts, which resolved in 2–3 weeks after DBBT session	0	-	-	0
Frances et al. 2014 [58]	0	-	-	0	-	-	0	-	-	Topical imiquimod 5%
Hata et al., et a. 2015 [59]	21	Surgical excision (no other details) Margin status: R+ in 16 cases (76.2%)	NR	16	Aim: adjuvant Volume 1: tumor bed Median total dose: 59.4 Gy (range, 45–648 Gy) in 30 fractions (range, 23–36 fractions) Volume 2: pelvic and inguinal lymph node area; total dose: 44–50. 4 Gy. In 3 cases (who had not undergone lymph node removal): total doses of 56, 60 and 612 Gy were delivered with the shrunken field to the enlarged inguinal lymph nodes of the three patients	Acute: 8 cases grade 1 leukopenia;16/16 ≤ grade 2 dermatitis, 9 grade 1 colitis 3 grade 1 cystitis no grade ≥ 3. Late telangiectasia grade 1	0	-	-	0
Asaka et al. 2015 [60]	1	Excisional procedures Wide local excision Margin status: NR	NR	0	-	-	0	-	-	-
Sopracordevole et al. 2016 [61]	25	WLE (5–20%) Partial SV (8–32%) Total SV (9–36%) Skinning (1–4%) Total vulvectomy (2–8%) Total vulvectomy with inguinofemoral lymphadenectomy (11–40.7%) Margin status: R0	NR	0	-	-	0	-	-	2: CO_2_ laser excision
Cowan et al. 2016 [62]	8	Unilateral PV (5), bilateral PV (1), multiple vulvectomies and skin flaps (1) Margin status: R1 microscopic	NR	0	-	-	0	-	-	0
Fan et al. 2016 [63]	18	10 local excision (55.5%) 3 WLE (16.7%) 5 SV (27.8%) Margin status: R1 in 3 cases (16.7%)	NR	0	-	-	0	-	-	0
Nagai et al. 2016 [64]	2	Wide local excision + split-thickness skin graft	Graft infection	0	-	-	0	-	-	0
Onaiwu et al. 2016 [65]	74	WLE (61.8%) RV (14.6%) Skinning vulvectomy (4.5%) Mohs surgery (2.3%) Margin status: R1 in 47 cases (63.5%)	NR	0	-	-	0	-	-	15
Hillmann et al. 2016 [66]	1	Radical vulvectomy and bilateral inguinal lymphadenectomy Margin status: R0	NR	0	-	-	0	-	-	0
Liau et al. 2016 [67]	2	1 WLE 1 Laser vulvectomy Margin status: NR	NR	0	-	-	0	-	-	3: topical imiquimod 5% cream
Jeon et al. 2016 [68]	1	Minimal excision with a 1 cm safety margin Margin status: R0	NR	1	1.8 Gy per day, 5 times per week, up to 50.4 Gy over 6 weeks to perineum, both inguinal lymph nodes, followed by 19.8 Gy to perineum for 2 weeks	Minimal excision with a 1-cm safety margin	0	-	-	0
Dogan et al. 2017 [69]	1	HV → WLE Margin status: R+ → R+	NR	0	-	-	0	-	-	1 (topical imiquimod)
Vicentini et al. 2017 [70]	0	-	-	0	-	-	0	-	-	1 (topical imiquimod, laser, photodynamic therapy
Parashurama et al. 2017 [71]	18	WLE (88.9%) Radical excision (11.1%) Margin status: R1 in 13 cases (72.2%)	NR	3	45 Gy in 25 fractions was delivered to the pelvis over 5 weeks with a boost to the vulva of 9 Gy in 5 fractions	NR	0	-	-	0
Konstantinova et al. 2017 [72]	102	WLE, local resections, or vulvectomies Margin status: NR	NR	0	-	-	0	-	-	0
Mota et al. 2017 [73]	1	SV Margin status: NR	NR	0	-	-	0	-	-	0
Long et al. 2017 [74]	90	WLE (96.7%) Abdominoperineal resection (3.3%) Margin status: R1 in 39.2%	NR	0	-	-	0	-	-	0
Kato et al. 2018 [75]	2	1: wide local excision of the primary site and biopsies of both inguinal lymph nodes 1: wide local excision and left groin lymph node dissection	NR	2	Aim: adjuvant at ilioinguinal areas and iliac region	NR	2	Aim: palliative Scheme: 1° line CT: docetaxel (60 mg/m^2^) 2° line CT: TS-1 (100 mg/body)	NR	0
Borghi et al. 2018 [76]	79	Vulvar resection, simple vulvectomy, radical vulvectomy. Locoregional (inguinal or pelvic) lymphadenectomy	NR	7	Aim: adjuvant	NR	7	Aim: palliative Scheme: 5-FU, cisplatin, carboplatin, mitomycin C, vincristine, etoposide, and docetaxel, either as single agents or in combined regimes	NR	0
Sawada et al. 2018 [77]	4	Simple vulvectomy	NR	0	-	-	0	-	-	9 (topical imiquimod)
Hiratsuka et al. 2018 [78]	0	-	-	1	Aim: radical Total dose/Fr: 25 Gy-Eq Technique: BNCT Volume: NR	Acute moderate skin erosion and dysuria	0	-	-	0
Hsieh et al. 2018 [79]	6	WLE Margin status: NR	NR	NR	Aim: rescue RT Total dose/Fr and technique: external beam vulvar radiation 56 Gy + interstitial brachytherapy to 30 Gy. Volume: perineum and the left-side inguinal nodes + interstitial brachytherapy	NR	NR	Aim: palliative Scheme: 1° line CT: 6 cycles of carboplatin AUC 5 + paclitaxel 175 mg/m2 q21, TZB from cycles 2 to 6 + TZB 5 cycles maintenance 2° line CT: T-DM1 3.6 mg/kg q21 for 6 cycles	2° line CT: G1 fatigue and bleeding, (intermittent vaginal bleeding, hematuria, epistaxis, and gum bleeding, which started after cycle 2)	0
Nitecki et al. 2018 [80]	42	WLE (19%) Partial SV (26.2%) RV (47.7%) Vulvar mapping (7.1%) Margin status: R+ in 3 cases (7.1%)	NR	0	-	-	0	-	-	0
Rioli et al. 2018 [81]	0	-	-	0	-	-	0	-	-	13
Kato et al. 2018 [82]	0	-		0	-	-	8	Aim: palliative Scheme: 5-fluorouracil (500 mg/body, 7 days/week) and cisplatin (5 mg/body 5 days/week)	-	0
Cai et al. 2018 [83]	0	-	-	0	-	-	8	Aim: palliative Scheme: docetaxel 60 mg/m day 1; cisplatin 25 mg/m day 1–3	-	0
Bouceiro-Mendes et al. 2019 [84]	1	Local surgical excision followed by WLE ➝ total Vu with fasciocutaneous flap reconstruction. Margin status: R1 (both excision); R0 (Vu)	NR	1	Aim: rescue RT at 4 years after surgery Total dose/Fr: 59.4 Gy Technique: NR Volume: NR	None	0	-	-	0
Loiacono et al. 2019 [85]	24	WLE (25%), SV (33.3%), extended Vu (41.7%) Margin status: R1 in 12 patients	Wound dehiscence (4) and 1 case urethral stenosis (1)	0	-	-	0	-	-	0
Mujukian et al. 2019 [86]	1	Wide local excision and then radical excision with a split skin graft reconstruction 6 years after the original diagnosis	NR	0	-	-	0	-	-	1 (topical imiquimod)
Molina et al. 2019 [87]	3	Local excision	NR	0	-	-	0	-	-	3 (topical imiquimod 5% in all, photodynamic therapy in case 1)
Hirai et al. 2019 [88]	0	-	-	0	-	-	5	CDDP (30 mg/m^2^), EPI (50 mg/m^2^), and TXL (120 mg/m^2^) per cycle	Ematotoxicity (100%)	0
van der Linden et al. 2019 [89]	74	ni-VPD (51): 64.7% WLE, 9.8% HV, 23.5% (skinning) Vu mi-VPD (10): NR i-VPD (13): 37.5% skinning Vu, 12.5% HV, 31.3% WLE, 29.2% groin surgery. Margin status: ni-VPD: R+ in 82.4% mi-VPD: R+ i-VPD: R+ in 76.9%	NR	3 (1 in mi-VPD group, 2 in i-VPD group)	NR	NR	1	In i-VPD group Aim: palliative Scheme: NR	NR	21
Panoskaltsis et al. 2019 [90]	1	WLE	NR	1	Aim: adjuvant	-	0	-	-	0
Nasioudis et al. 2020 [91]	2412	WLE (46.9%) PV (34.1%) RV (4.2%) Total vulvectomy (6.4%) NR (6.8%) Margin status: R+ in 52.7%	NR	35	NR	NR	0	-	-	121 (immunotherapy)
Bartoletti et al. 2020 [92]	4	Case 1: vulvo-perianal WLE with left colostomy ➝ ileo-inguinal LA and enlargement of previous perineal resection Case 2: SV ➝ vulvar resection ➝ urethral relapse excision ➝ new resection of the urethral meatus Case 3: SV with positive margins ➝ RV with the excision of the perineum, perianus, and part of the gluteal muscle ➝ bulky inguinofemoral LNs excision Case 4: laser vaporization of the anal and perianal mucosa and a SV with skin grafts, with two large triangular flaps bilaterally (V-Y plastic) ➝ skinning Vu of the anterior hemi-vulva with plastic reconstruction and perianal/anal excision Margin Status: NR	NR	2	Aim: palliative RT after surgery Total dose/Fr: Case 2: 30 Gy Case 3: 30 Gy/10 fr Technique: Case 2: NR Case 3: whole brain radiation Volume: Case 2: pelvis Case 3: brain	NR	4	Aim: palliative after surgery Scheme: Case 1: w-paclitaxel + w-TZB for 9 months, then maintenance with 3-w-TZB for 1 year ➝ re-challenge with w-paclitaxel and TZB for 7 months, then maintenance with 3-w-TZB for 7 months ➝ TZB emtansine for 2 months, ongoing Case 2: 3-w-cisplatin for 2 cycles➝ w-paclitaxel + w-TZB for 5 months Case 3: w-paclitaxel for 4 total cycles + w-TZB for 4 months, then 3-w-TZB for 11 months ➝ 3-w-TZB and carboplatin for 1 cycle Case 4: 3-w-TZB ➝ w-paclitaxel and 3-w-TZB	Case 1: G3: neutropenia Case 2: G2: anemia, onychopathyCase 3: G1: fatigue, constipation, nausea, anemia and alopecia, G2: neurotoxicityCase 4: G2: neurotoxicity	3
Bruce et al. 2020 [93]	1	Left RV ➝ bilateral sentinel LN biopsy ➝ left inguinal LA. Margin status: R0 in adenocarcinoma area, R+ in EMPD area	NR	1	Aim: adjuvant Total dose/fr: 5940 cGy/33 fr Technique: NR, but in combination with CT Volume: pelvis	NR	1	Aim: adjuvant Scheme: 2 cycles of carboplatin + paclitaxel ➝ delivery ➝ CT-RT with w- cisplatin ➝ TZB 1 year	NR	0
Noel et al. 2020 [94]	1	Left RV with left groin dissection, laparoscopic left inguinal LN dissection, and perineal V-Y plasty repair Margin status: NR	NR	1	Aim: adjuvant Total dose/Fr: NR Technique: NR Volume: NR	NR	0	-	-	0
Kilts et al. 2020 [95]	1034	Surgery	NR	0	-	-	0	-	-	0
Rathore et al. 2020 [96]	0	-	-	0	-	-	0	-	-	1
Sarkar et al. 2020 [97]	1	WLE Margin status: R0 inferior, R+ superior and lateral	NR	1	Aim: adjuvant Total dose/fr: 50 Gy/28 fr Technique: NR Volume: groin, bilateral inguino-femoral and pelvic LNs	NR	0	-	-	0
Sopracordevole et al. 2020 [98]	7	WLE (57%), superficial total Vu (29%), skinning Vu (14%) Margin status: R+ in 3 cases (42.9%)	NR	3	Aim: prior treatment Total dose/Fr: NR Technique: NR Volume: NR	NR	0	-	-	0
Stasenko et al. 2020 [99]	19	NR	NR	0	-	-	1	Aim: palliative Scheme: *PIK3CA* inhibitor in clinical trial	NR	7
Liang et al. 2021 [100]	38	ni-VPD (29): PV (48%), SV (28%), RV (14%), WLE (10%) mi-VPD (8): PV (88%), SV (12%) i-VPD (1): NR Margin status: ni-VPD: R+ in 51.7% mi-VPD: R+ in 12.5% i-VPD: NR	NR	1	Aim: adjuvant, concurrent CT-RT Total dose/Fr: NR Technique: NR Volume: NR	NR	1	Aim: adjuvant, concurrent CT-RT Scheme: NR	NR	0
Hirata et al. 2021 [101]	1	Surgical resection of vulvar and perineal tumors Margin status: NR	NR	0	-	-	1	Aim: palliative Scheme: docetaxel, for more than 16 months	NR	0
Kosmidis et al. 2021 [102]	2	WLE Margin status: R0	None	0	-	-	0	-	-	0
Mazzilli et al. 2021 [103]	1	NR	NR	-	-	-	0	-	-	0
Preti et al. 2021 [104]	95	WLE (41–44%) HV (26–27%) TV (13–16%) Margin status: R+ in 91.7%	NR	0	-	-	0	–	-	27
Liu et al. 2021 [105]	54	25 extended resection (8 experienced unilateral or bilateral inguinal lymph node metastasis and underwent inguinal lymph node dissection), 29 palliative resection Margin status: NR	NR	0	-	-	0	-	-	0
Ferrara et al. 2021 [106]	0	-	-	0	-	-	0	-	-	10
Bajracharya et al. 2022 [107]	1	RV + bilateral groin dissection + bilateral gracilis pedicles flap + urethral and introitus reconstruction Margin status: R0	NR	1	Aim: adjuvant Total dose/Fr: NR Technique: NR Volume: NR	NR	0	-	-	0
Wang et al. 2022 [108]	0	-	-	0	-	-	0	-	-	2
Borella et al. 2022 [109]	0	-	-	0	-	-	0	-	-	55
Van der Linden et al. 2022 [110]	0	-	-	0	-	-	0	-	-	24

➝ = for relapse; AWD = alive with disease; Cr = case report; CR = complete response; Cs = case series; CT = chemotherapy; DCR = disease control rate; DID = died of intercurrent disease; DOD = died of disease; HV = hemivulvectomy; i-VPD = invasive vulvar Paget’s disease; LA = lymphadenectomy; LFU = lost to follow-up; LN = lymph node; MAL-PDT = methyl 5-aminolevulinic photodynamic therapy; mi-VPD = microinvasive vulvar Paget’s disease; NED = no evidence of disease; ni-VPD = noninvasive vulvar Paget’s disease; NR = not reported; PD = Progressive disease; PPS = prospective pilot study; PR = partial response; pts = patients; PV = simple partial vulvectomy; R = residual disease; Ra = retrospective analysis; RT = radiotherapy; RV = radical vulvectomy; SV = simple vulvectomy; TZB = trastuzumab; Vu = vulvectomy; w = weekly; WLE = wide local excision, 5-FU = 5-fluorouracil.

**Table 3 cancers-15-01803-t003:** Outcomes and recurrence treatments.

Reference	Median FU	DCR/Recurrence Rate	PFS	OS	Recurrence Treatment	
					Systemic PD	Local Recurrence
Kodama et al. 1995 [16]	NR	NR	ni-VPD + mi-VPD: 65.7% i-VPD: 27.7%	NR	i-VPD: CT, RT	ni-VPD: CT, RT, surgery mi-VPD: surgery i-VPD: CT, RT, surgery
Fishman et al. 1995 [17]	NR	Mean time to recurrence: negative margins: 4.4 (3–12) years positive margins: 1.4 (1–11.8) years Recurrence rate: 35.7%	NR	NR	-	Surgery
Yoshitatsu et al. 1997 [18]	6 years	CR	6 years	6 years	-	Surgery
Goldblum et al. 1997 [19]	9 years	ni-VPD: 4 pts with local recurrence from 25 months to 12 years i-VPD: 1 died of the disease	NR	NR	-	Surgery
Fanning et al. 1999 [20]	7 years (3–23)	Recurrence rate: 34% at a median of 3 (0.6–15) years	3 years	NR	-	-
Henta et al. 1999 [21]	NR	Nearly local CR	NR	NR	-	-
Murata et al. 1999 [22]	NR	PR to RT → died of the disease	NR	13 months	-	-
Louis-Sylvestre et al. 2001 [23]	5.1 years (1–12.1)	Mean time to recurrence: After laser alone: 1 ± 0.6 years After limited excision + peripheral laser: 1.9 ± 1.5 years After wide excision alone: 2.7 ± 1 years Recurrence rates at 1 year: After laser alone: 67% After limited excision + peripheral laser: 33% After wide excision alone: 23%	NR	NR	-	-
Wilkinson et al. 2002 [6]	NR	NR	Case 1: NR Case 2: no PD at 9 months Case 3: NR	Case 1: NR Case 2: still alive Case 3: NR	-	-
Tebes et al. 2002 [24]	13.5 months (1–216)	Mean time to recurrence: 30 months (3–89)	NR	NR	-	Surgery
Luk et al. 2003 [25]	NR	NR	10 months	15 months	-	-
Wang et al. 2003 [26]	NR	CR	NR	NR	-	Imiquimod
Chin et al. 2004 [27]	NR	-	8 years	-	-	-
Zawislak et al. 2004 [28]	3 months	CR	-	-	-	-
Bhattacharya et al. 2005 [29]	8 years	NR	NR	NR	-	Surgery
Raspagliesi et al. 2006 [30]	1–5 months	DCR: 100% (CR 57.1%, PR 42.9%)	NR	NR	-	-
Yanagi et al. 2007 [31]	5 months (case 1) and 2 years (case 2)	CR	NR	NR	-	-
Hatch et al. 2008 [32]	12 months 6 months	CR	NR	NR	-	-
Karam et al. 2008 [33]	NR	NR	NR	NR	-	Imiquimod, TZB
Challenor et al. 2009 [34]	4 months 3 months	CR	NR	3.5 months	-	-
Sendagorta et al. 2010 [35]	22 months (20–26)	CR	NR	NR	-	-
Shaco-Levy et al. 2010 [36]	NR	Recurrence rate: 32%	NR	43% NED, 43% DID, 3% LFU, 2% DOD with invasive adenocarcinoma, 9% AWD	-	-
Roh et al. 2010 [37]	36.6	RR: 56%	NR	NR	NR	NR
Anton et al. 2011 [38]	NR	CR	NR	NR	-	-
Feldmeyer et al. 2011 [39]	12 months	CR	12 months	12 months	-	-
Hanawa et al. 2011 [40]	NR	PR after 4 courses of CT, then PD	NR	NR	RT in inguinal LNs and primary lesion	RT
Jones et al. 2011 [41]	NR	NR	NR	NR	NR	NR
Tonguc et al. 2011 [42]	24 months	CR	24 months	24 months	-	Surgery, imiquimod
Mendivil et al. 2012 [43]	53 months	RR: 56.2%	NR	NR	NR	NR
Al Yousef et al. 2012 [44]	NR	Persistence of disease: 100%	NR	NR	NR	NR
Baiocchi et al. 2012 [45]	30.5 months (21–40)	DCR: 100% (PR: 25%; CR: 75%) Recurrence rate: 25%	NR	30.5 months	-	-
Wakabayashi et al. 2012 [46]	NR	Recurrence	5 years after 1° surgery, 3 years after 2° surgery	NR	3-w-TZB	-
Cai et al. 2013 [47]	NR	Recurrence rate: 34.3%	NR	Intraepithelial cases: 24.5 months Invasive cases: 70.8 months Cases with adnexal adenocarcinoma: 21.3 months	Surgery (12) RT (3)	-
Choi et al. 2013 [48]	38 months (34–46)	CR: 100%	38 months	38 months	-	-
Gavriilidis et al. 2013 [49]	7 months	NR	NED	NED	-	-
Sanderson et al. 2013 [50]	18 months (12–24)	CR: 50% PD: 33.3% Recurrence rate: 16.6%	NR	18 months	-	-
De Magnis et al. 2013 [51]	76.9 months	Recurrence rate: 44.1%	45.7 months	NR	-	WLE
Treglia et al. 2013 [52]	NR	DCR: 3 years	3 years	3 years	Chemotherapy	Chemotherapy
Magnano et al. 2013 [53]	NR	DCR: 2 months	NR	NR	MAL-PDT (3 treatments) + topical tretinoin 0.05% and vitamin E with CR at time of writing (6 months)	
Marchitelli et al. 2014 [54]	18 months	CR: 90%	NR	14.4 months	-	-
Liu et al. 2014 [55]	43.6 months	Recurrence rate: 43.5%	12.7 months	NR	-	-
Luyten et al. 2014 [56]	14.4 months	NR	NR	14.4 months	-	-
Carrozzo et al. 2014 [57]	34 months	All patients showed complete remission of the disease without histological evidence of tumor at the end of the treatments *	34 months	34 months	* Second course of brachytherapy in persistent/recurrent disease	-
Frances et al. 2014 [58]	NR	CR at 2 weeks after the end of the treatments	NR	NR	-	-
Hata et al. 2015 [59]	38 months (range, 2–109 months)	3- and 5-year overall local control: 100%	3-year distant metastasis-free rates: 66% 5-year distant metastasis-free rates: 55%	3-year OS: 92%; 5-year OS: 62% 3-year cause-specific survival rates: 92% 5-year cause-specific survival rates: 71%	-	-
Asaka et al. 2015 [60]	32 months	CR	NR	Still alive	-	-
Sopracordevole et al. 2016 [61]	79.5 months	Recurrence rate: 29.6%	NR	NR	-	-
Cowan et al. 2016 [62]	35 months	DCR: 75% (CR) Of the 6 pts for whom a CR was achieved, VPD recurred in 4 (67%)	NR	NR	-	-
Fan et al. 2016 [63]	70 months	Recurrence rate 11%	NR	NR	-	-
Nagai et al. 2016 [64]		Recurrence rate: 50%	8 months	25 months	-	-
Onaiwu et al. 2016 [65]	73.9 months	Recurrence rate: 58.4%	NR	73.2 months	-	Excision + local imiquimod
Hillmann et al. 2016 [66]	NR	NR	NR	NR	-	-
Liau et al. 2016 [67]	NR	NR	NR	NR	-	Imiquimod treatment Total duration of the treatment: 22-100-16 months
Jeon et al. 2016 [68]	NR	NR	NR	NR	-	
Dogan et al. 2017 [69]	6 months	CR	6 months	6 months	-	-
Vicentini et al. 2017 [70]	NR	NR	NR	NR	-	-
Parashurama et al. 2017 [71]	NR	Recurrence rate: 67%	NR	8 years	-	-
Konstantinova et al. 2017 [72]	NR	NR	NR	NR	-	-
Mota et al. 2017 [73]	NR	NR	NR	NR	-	-
Long et al. 2017 [74]	5 years	Recurrence rate: 37.7%	56.1% at 5 years	NR	-	-
Kato et al. 2018 [75]	5.7 months 11 months	18 weeks after surgery 10 months after surgery	NR	5.7 months 11 months	-	-
Borghi et al. 2018 [76]	57 months	Recurrence rate: 60%, with a mean time to first recurrence of 20 (range, 5–36) months	NR	Invasive group: 70% at 48 months	-	-
Sawada et al. 2018 [77]	46 months	DCR: 100% (56% CR, 44% PR)	NR	NR	-	-
Hiratsuka et al. 2018 [78]	NR	DCR: CR in 6 months	3.2 years	3.2 years (DID)	-	-
Hsieh et al. 2018 [79]	NR	DCR: CR after 1° line CT, CR after 6 T-DM1 cycles	7.0 months with 1° line CT followed by TZB maintenance; 6.0 months with T-DM1	NR	TDM1	6 excisional procedures, 3 laser ablations, several courses of topical imiquimod treatment, RT
Nitecki et al. 2018 [80]	45.8 months	Recurrence rate: 2 months (1–6)	28.7 months (0–169)	NR	-	Weekly cisplatin + palliative radiation
Rioli et al. 2018 [81]	38 months	Recurrence rate: 7%	NR	NR	-	-
Kato et al. 2018 [82]	-	DCR: 75% (50% PR, 25% SD) PD: 25%	25.0 weeks	77.4 weeks	-	-
Cai et al. 2018 [83]	53 months	DCR: 100% (50% PR, 50% SD)	9.9 months	28.9 months	-	-
Bouceiro-Mendes et al. 2019 [84]	4 years	Disease control: 4 years post-Vu; 3 months post-RT	4 years	4 years	-	RT 4 years after surgery; imiquimod after 3 months post-RT
Loiacono et al. 2019 [85]	NR	Recurrence rate: 33%	NR	NR	-	Surgery
Mujukian et al. 2019 [86]	NR	NR	NR	NR	-	Surgery
Molina et al. 2019 [87]	NR	NR	NR	NR	-	-
Hirai et al. 2019 [88]	NR	-	20.1 months	8 months	-	-
van der Linden et al. 2019 [89]	38 months	Recurrence rate: 36.4%	ni-VPD: 69.3 months i-VPD: 26.5 months	NR	CT	-
Panoskaltsis et al. 2019 [90]	18 months	CR	NR	18 months	-	-
Nasioudis et al. 2020 [91]	66.5 months	NR	NR	NR	-	-
Bartoletti et al. 2020 [92]	NR	NR	NR	NR	CT	Surgery
Bruce et al. 2020 [93]	NR	NR	NR	NR	-	-
Noel et al. 2020 [94]	NR	NR	NR	NR	-	-
Kilts et al. 2020 [95]	NR	NR	NR	Five-year cancer specific survival (CSS) was 95.5% and was associated with the stage. Compared to patients with localized disease, patients with distant metastases had dramatically worse CSS (HR: 85.8 (31.8–248) *p* < 0.0001). Vital status (120 months): alive, 569 (94.7%); died, 32 (5.3%)	-	-
Rathore et al. 2020 [96]	NR	NR	1 month	1 month	-	-
Sarkar et al. 2020 [97]	NR	DCR: 18 months	18 months	Still alive	-	-
Sopracordevole et al. 2020 [98]	NR	DCR: 24 months	NR	NR	-	Surgery (9 pts, 90%); third treatment for recurrence (4 pts, 40%), a quarter treatment (2 pts, 20%), fifth treatment for recurrence of disease (1 pt, 10%)
Stasenko et al. 2020 [99]	8 months	Recurrence rate: 62%	NR	NR	PIK3CAi (1 pt 6%)	Surgery (10 pts, 63%), imiquimod (4 pts, 25%), topical 5-FU (1 pt 6%).
Liang et al. 2021 [100]	NR	NR	NR	NR	-	Surgical excision
Hirata et al. 2021 [101]	NR	NR	NR	NR	-	-
Kosmidis et al. 2021 [102]	NR	NR	NR	NR	-	-
Mazzilli et al. 2021 [103]	NR	NR	NR	NR	-	-
Preti et al. 2021 [104]	94.6 months	Recurrence rate: 73%	NR	39 months	-	-
Liu et al. 2021 [105]	2 years and at most 10 years	NR	NR	NR	-	-
Ferrara et al. 2021 [106]	12 months	Recurrence rate: 60%	NR	12 months	-	-
Bajracharya et al. 2022 [107]	24 months	-	24 months	24 months	-	-
Wang et al. 2022 [108]	17.4 months	Recurrence rate: 36.4% DCR: Case 1: relapse at 24 month (FU 27 months). Case 2: no relapse (FU 14 months)	24 months	NR	-	-
Borella et al. 2022 [109]	66 months	NR	NR	31 months	-	-
Van der Linden et al. 2022 [110]	31 months	Recurrence rate: 34.8%	NR	31 months	-	-

➝ = for relapse; AWD = alive with disease; Cr = case report; CR = complete response; Cs = case series; CT = chemotherapy; DCR = disease control rate; DID = died of intercurrent disease; DOD = died of disease; HV = hemivulvectomy; i-VPD = invasive vulvar Page’s disease; LA = lymphadenectomy; LFU = lost to follow-up; LN = lymph node; MAL-PDT = methyl 5-aminolevulinic photodynamic therapy; mi-VPD = microinvasive vulvar Paget’s disease; NED = no evidence of disease; ni-VPD = noninvasive vulvar Paget’s disease; NR = not reported; PD = Progressive disease; PPS = prospective pilot study; PR = partial response; pts = patients; PV = simple partial vulvectomy; R = residual disease; Ra = retrospective analysis; RT = radiotherapy; RV = radical vulvectomy; SD, stable disease; SV = simple vulvectomy; TZB = trastuzumab; Vu = vulvectomy; w = weekly; WLE = wide local excision.

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
