# Peer review of "Vulvar Paget’s Disease: A Systematic Review of the MITO Rare Cancer Group"

_cancers, 2023, doi:10.3390/cancers15061803_

Round 1

Reviewer 1 Report

Thank you for the opportunity to review this interesting paper.

The topic is debated in literature, indeed a clear indication to medical or surgical treatment is lacking. Nevertheless, the manuscript is well written

Here is my concerns:
- a recent paper on this topic has been published, and it is very detailed. I advise to take it into account (https://doi.org/10.3390/jpm13010100).

- I would do a comparison with vulvar cancer, in particular when you talk about in discussion section of hormone receptors status, HER2 etc (doi: 10.1007/s00432-020-03226-6; doi: 10.3390/cancers13246373)

- the Table is well organized. I would add a figure on flow chart for the choice of papers included in this study.

Reviewer 2 Report

Dear Authors 

the submitted paper is a complete and updated overview on an interesting topic. Introduction, Methods, Discussion and Conclusion are well reported.
I would suggest to review the report of results in Table 1, since the overview of the 29 included studies is not easily available in the current layout. I would suggest to change the layout, to reduce the information in Table or to divide Table into two, to give the reader a better vision of the included studies.
I would also suggest authors to include observation about the correct preoperative diagnostic workup by including reference by Garganese et al entitled " "Clock mapping" prior to excisional surgery in vulvar Paget's disease: tailoring the surgical plan” published in Arch Gynecol Obstet in 2022. Some other information about invasive forms are included in the review entitled "Invasive Paget Disease of the vulva" by Borghi et al, published in Int J Gynecol Cancer in 2018. These could be good citations.

Reviewer 3 Report

The paper would be a review of Paget disease of the vulva. 

the main weakness of the paper is the lack of novelty. It is a mere description of some studies. It does not add anything new. 

The language needs extensive editing. (I suggest by a native speaker)

I suggest performing a systematic review using PRISMA guidelines. 

The readability is poor and content should be reported in a clearer fashion. 

Reviewer 4 Report

The manuscript "Vulvar Paget’s Disease: A Review of the MITO Rare Cancer Group" from Giuseppe Caruso et al. is a interesting and nicely written systematic review of the literatue summarizing the current treatment options for VPD. VPD is a very rare disease with inhomogenous clinical studies, case reports and individual treatment summaries. this increases the necessity for such systematic review of the current literature. 

As a minor point: The manuscript could further be improved if the authors could draw an outlook into the future therapy of VPD within the discussion. Will invasive surgery and / or thermal laser ablation persist as gold standard? Could a local treatment with physical plasma gas be an option for future VPD treatment as recently shown by a working group in Tübingen in case of cervical intraepithelial neoplasia? (Marzi et al. Cancers 2022)

Minor revision: Table 1 definitely needs to be improved. Perhaps the authors could increase the clarity by displaying the table in landscape format instead of portrait format.  

Round 2

Reviewer 3 Report

The authors are not providening any relevant clinical message. 

The paper is very difficult to be followed. The language needs substantial improvements. Readibility is one the main weakneses of this manuscript. 
